# Validation and adaptation of the Arabic version of health-related quality of life with six domains (HRQ-6D): A factor and Rasch analyses study

**Walid Al-Qerem**[1]*, **Anan Jarab**[2,3], **Abdel Qader Al Bawab**[1], **Judith Eberhardt**[4], **Fawaz Alasmari**[5], **Alaa Hammad**[1], **Lujain Al-Sa'di**[1], **Raghd Obidat**[1], **Sarah Abu Hour**[1], **Taha Al-Hayali**[1]

**1** Department of Pharmacy, Faculty of Pharmacy, Al-Zaytoonah University of Jordan, Amman, Jordan, **2** College of Pharmacy, Al Ain University, Abu Dhabi, United Arab Emirates, **3** Department of Clinical Pharmacy, Faculty of Pharmacy, Jordan University of Science and Technology, Irbid, Jordan, **4** Department of Psychology, School of Social Sciences, Humanities and Law, Teesside University, Middlesbrough, United Kingdom, **5** Department of Pharmacology and Toxicology, College of Pharmacy, King Saud University, Riyadh, Saudi Arabia

* waleed.qirim@zuj.edu.jo

## Abstract

### Background

Health-related quality of life (HRQOL) provides a complete picture of patients' overall health status and should be evaluated in all patients encountered. To accurately assess patients' HRQOL a comprehensive validated tool is necessary. Therefore, the present study aimed to validate the Arabic version of the Health-Related Quality of Life with Six Domains (Ar-HRQ-6D) scale.

### Methods

This cross-sectional study utilized an online questionnaire targeting adult Jordanians and implemented several validation steps to ensure the adequacy of the Ar-HRQ-6D. These steps included the application of the forward-backward translation technique, assessment of the content and face validity of the questionnaire, evaluation of internal consistency, validation of the construct through confirmatory factor analysis (CFA) and Rasch analysis, and assessment of the questionnaire's predictive capabilities.

### Results

A total of 808 participants (63% female) completed the Ar-HRQ-6D. Confirmatory Factor Analysis (CFA) supported the suitability of the original three-factor model for the present study data, yielding acceptable model fit indices ($\chi^2$/df = 4.1, SRMR = 0.03, RMSEA = 0.06, CFI = 0.99, GFI = 0.96, CIF = 0.98, and TLI = 0.97) and factor loadings ranging from 0.63 to 0.86. Cronbach's alpha for the three factors ranged from 0.81 to 0.90, confirming the high reliability of the questionnaire. Rasch analysis further validated the

**Data availability statement:** The dataset supporting the conclusions of this article is available in the Zenodo repository https://doi.org/10.5281/zenodo.13929837.

**Funding:** The current project was funded by Al-Zaytoonah University of Jordan, Ref no: 10/4/2023-2024. The funders had no role in study design, data collection and analysis, decision to publish, or preparation of the manuscript.

**Competing interests:** The authors have declared that no competing interests exist.

person and item separation reliability for the three factors. Additionally, all items of the Ar-HRQ-6D fell within the acceptable infit and outfit ranges. All thresholds were appropriately ordered, ranging from -5.27 to 2.86. Significant differences were observed in the median Ar-HRQ-6D scores across the various health status categories (p < 0.001), with the healthy category showing significantly higher median scores than the other categories. These findings confirm the predictive validity of the Ar-HRQ-6D.

## Conclusion

The study confirmed the reliability, validity, and predictive accuracy of the Arabic version of the Ar-HRQ-6D. This tool is suitable for assessing patients' HRQOL across various medical settings.

## Introduction

Health-related quality of life (HRQOL) is "a multidomain concept that represents the patient's general perception of the effect of illness and treatment on physical, psychological, and social aspects of life" [1]. It refers to a statistical index encompassing various parameters including health-related, economic, and environmental factors that influence people's living conditions [2]. Measuring HRQOL is essential for assessing healthcare practices and interventions. Furthermore, the significance of HRQOL is rising in the healthcare sector, particularly in light of the increasing complexity, capabilities, and costs of modern medical treatments [3]. Several instruments have been developed and are used to document the burden associated with specific diseases, assess the effectiveness of new technologies or interventions, and inform or support health policy [3].

Various HRQOL assessment indicators provide a thorough evaluation of the burden posed by disabilities, injuries, and preventable diseases [4]. HRQOL indicators are patient-centric assessments, as they capture patients' perspectives on their treatment and illness, their outcome preferences, and their perceived healthcare needs [5]. HRQOL comprises four main dimensions: mental state, economic and social conditions, physical and motor skills, and somatic perception (e.g., pain) [2].

In Jordan, healthcare services are delivered through a combination of public, private, and military institutions. The country has 122 hospitals, 70 of which are private, and a total of 16,057 hospital beds, with 51% in public hospitals. Approximately 9% of Jordan's GDP is allocated to healthcare, reflecting the country's emphasis on primary care and preventive services [6]. However, there remains a growing need for culturally appropriate tools to evaluate HRQOL, particularly in managing chronic diseases, which account for 76% of total deaths. Diabetes alone affects 34% of Jordanians aged 25 and older [7].

Multiple quantitative studies have been conducted to provide statistical insight into HRQOL in different countries. For example, a cross-sectional study reported that colorectal cancer had an adverse impact on the HRQOL of patients [8]. Similarly, a cross-sectional study of heart failure patients in Jordan found that physical symptoms such as edema, dyspnea, activity intolerance, and fatigue significantly contributed to a decline in HRQOL [9]. Furthermore, a study conducted with diabetic patients in Jordan, revealed that insulin administration, low-income status, marital status, and presence of diabetic complications significantly influenced their quality of life [10].

One of the most widely used tools for assessing HRQOL is the European Quality of Life 5 Dimensions 3 Level Version (EQ-5D-3L), a standardized tool that measures an

individual's overall health status [11]. The EQ-5D-3L consists of two parts: a descriptive section and an evaluation of the individual's overall health status. The descriptive part covers five main domains: mobility, usual activities, self-care, anxiety/depression, and pain/discomfort. The second part assesses overall health using a visual analogue scale (EQ-VAS) ranging from zero (representing the worst imaginable health) to 100 (representing the best imaginable health) [12]. Nevertheless, various studies have recommended the replacement of the 3-level response scale with a 5-level response scale to improve the accuracy of evaluating specific health outcomes; thus, the 5-level version of the scale (EQ-5D-5L) was developed [13,14]. However, a major drawback of both the EQ-5D-5L and EQ-5D-3L is that each domain is measured by a single item only, which makes it impossible to assess the internal consistency of each domain and casts doubt on its reliability. Furthermore, the validation of the EQ-5D relied solely on expert panel opinions and comparisons of results from individuals with different health statuses [15,16]. However, it did not incorporate more advanced validation techniques, such as the Rasch model, exploratory factor analysis (EFA), or confirmatory factor analysis (CFA). Moreover, the content validity of the EQ-5D could be improved by including an additional domain to assess an individual's current health status [17]. To address these limitations, the newly developed Health-Related Quality of Life with Six Domains (HRQ-6D) [17] was specifically designed.

The present study sought to address the need for a comprehensive and culturally relevant tool to assess HRQOL in Arabic-speaking populations. The HRQ-6D was selected for its ability to evaluate a broader range of HRQOL dimensions with enhanced accuracy. This study aimed to validate the Arabic version of the HRQ-6D (Ar-HRQ-6D), ensuring it is both reliable and culturally appropriate for use in Arabic-speaking healthcare settings.

## Materials and methods

### Study design and participants

In this cross-sectional study, an online questionnaire was distributed using Google Forms via multiple Jordanian social media platforms including WhatsApp, Facebook and LinkedIn, in the period between June and September 2024. The HRQ-6D questionnaire was selected after a review of relevant studies in the literature [17]. The inclusion criteria required participants to be Jordanian residents aged 18 years or older. Accordingly, screening questions to verify that participants met these criteria were included at the beginning of the questionnaire. An introductory section outlined the aims of the study, assured participants of confidentiality, and anonymity, and emphasized the voluntary nature of participation. Individuals who agreed to participate were required to select the following options: "I have read the study information" and "I agree to participate and consent to the publication of all collected data." Selecting these options constituted informed consent to participate and to publish the results before proceeding to the questionnaire. The Google form was designed so that the participant had to complete all the mandatory questions in the questionnaire before submitting the form to eliminate missing data. The study adhered to the ethical guidelines set forth in the Declaration of Helsinki. Ethical approval was granted by Al-Zaytoonah University of Jordan on the 10th of April 2024 (Ref#10/4/2023-2024).

### Sample size calculation

When conducting factor analysis, it is recommended to determine the sample size using a participant-to-item ratio, with a maximum suggested ratio of 20:1 [18]. Given that the questionnaire consisted of 12 items, the minimum required sample size was 240 participants.

## Data collection and study instruments

Convenience and snowball sampling were used to recruit participants. The questionnaire consisted of four sections. The first one focused on participants' sociodemographic characteristics including gender, age, monthly income, educational level, marital status, residence (rural or urban), governorate of residence, smoking status, physical activity level, self-rated eating habits, health insurance coverage, and the presence of comorbidities. If participants indicated they had no comorbidities, they were automatically redirected to the third section, skipping the second section. The second section aimed to identify the specific comorbidities participants had. The third section evaluated participants' health status, offering five categories, from which participants selected the one that best described their current health. The final section included the Ar-HRQ-6D items, which assessed how participants felt about their health condition on the day they completed the questionnaire. Participants were asked to indicate the extent to which certain events had occurred or were relevant to their current health condition.

The Ar-HRQ-6D tool consists of six domains, with each domain containing two items, resulting in a total of twelve items. Responses were measured on a five-point Likert scale: 'strongly disagree,' 'disagree,' 'neutral,' 'agree,' and 'strongly agree.' The scoring mechanism was as follows: five points for 'strongly disagree,' four points for 'disagree,' three points for 'neutral,' two points for 'agree,' and one point for 'strongly agree.' A higher overall score indicated better HRQOL.

## Tool validation

A group of experts, including two clinical pharmacists and two public health specialists, assessed the content validity of the developed questionnaire [19]. Initially, two independent translators produced two Arabic versions of the questionnaire. These versions were then compared by the translators and researchers, resulting in a single initial Arabic draft. This draft was subsequently back-translated into English by two independent professional translators. The researchers and translators compared the two back-translated versions, the original English version, and the initial Arabic draft to create the final Arabic version. This thorough process ensured that the original meaning was preserved while also ensuring the cultural appropriateness of the Arabic version. A pilot study was conducted with 30 randomly selected Jordanian participants (17 of whom were female) to ensure the clarity and suitability of the questionnaire for the target population. The median age of the 30 participants was 27 years. The researchers approached participants at various pharmacies across different locations in Jordan. Participants confirmed that the questions and response options were clear, relevant, and appropriate and that completing the questionnaire required approximately 15 minutes. The data from the pilot study was not included in the final statistical analysis. Cronbach's alpha was used to assess the reliability and internal consistency of the tool. Confirmatory factor analysis (CFA) and Rasch analysis were performed to assess the construct validity of the questionnaire, and its ability to differentiate between participants' QoL.

## Statistical analysis

Statistical analyses were conducted using the Statistical Package for the Social Sciences (SPSS) version 26, Jamovi version 2.3.28, and R-studio version 2024.04.2 with the Test Analysis Modules (TAM). The continuous variables were presented as medians (with interquartile ranges), while categorical variables were presented as frequencies and percentages. CFA was used to assess the fit of the study data to the original construct of the questionnaire. Several indices were calculated to confirm construct validity, including the Comparative Fit Index (CFI), Goodness-of-Fit Index (GFI), minimum discrepancy ($\chi2/df$), Root Mean Square Error of

Approximation (RMSEA), Standardized Root Mean Squared Residual (SRMR), and Tucker-Lewis Index (TLI). Factor loadings were produced and assessed, and the internal consistency of each factor was evaluated by computing Cronbach's alpha values.

A Likelihood Ratio Test was conducted to determine the most suitable Rasch analysis approach: the Partial Credit Model (PCM) or the Rating Scale Model. Unlike the Rating Scale Model, the PCM does not assume equal distances between answer thresholds. Significant Likelihood Ratio results indicated the use of PCM [20] A multi-factorial Rasch analysis was then performed to assess the suitability of the tool. Person and item separation reliability were calculated, and infit/outfit statistics were produced, with acceptable mean square values (MSQ) for infit and outfit ranging between 0.6 and 1.4 [21]. Additionally, item locations and thresholds were generated, and the Wright map was examined. Differential Item Functioning (DIF) analysis between genders was performed. The predictive validity of the Ar-HRQ-6D was assessed by comparing the Ar-HRQ-6D median scores across different health status categories [17] using the Independent-Samples Median Test (ISMT). A difference of ≥ 0.43 logits on the logit scale indicated no significant difference between the two genders [22].

## Results

### Sample characteristics

Table 1 displays participants' socio-demographic profiles. A total of 852 individuals completed the questionnaire. Of these, 44 were excluded for not meeting the inclusion criteria (15 were under the age of 18, and 29 were not residents of Jordan). The final analysis included data from 808 participants (see Fig 1), of whom 63.5% were female. The median age was 24 (IQR: 21-36) years. Most participants were not married (69.8%), had a bachelor's degree (62%), earned less than 500 Jordanian Dinars (JOD) per month (56.2%), had never smoked (75.1%), had a sedentary lifestyle (62.6%), and were classified as healthy and free of comorbidities (64.4%).

Participants' responses to the Ar-HRQ-6D items are shown in Table 2. For the health dimension items, most participants responded with "agree" or "strongly agree." The most agreed/strongly agreed-upon response was "I feel a lack of physical energy" (59%), while the least frequent was "I feel depressed" (38.6%).

Additionally, a clear pattern of "disagree" or "strongly disagree" responses was observed for the body function and perception items. The most frequent disagreement was for "I have a problem attending to my self-care needs" (63.3%), while the least frequent was for "I am worried that I will suffer poor health within 5 years" (48.3%).

### Confirmatory factor analysis

CFA was performed to evaluate the 3-factor model proposed in the original questionnaire. The results indicated that the distribution of the 12 items across the three dimensions was suitable for the present study. The model yielded acceptable fit indices ($\chi 2/df$ = 4.1, SRMR = 0.03, RMSEA = 0.06 CFI = 0.99, GFI = 0.96, CIF = 0.98, and TLI = 0.97). Table 4 presents the standardized factor loadings, standard error for each item, and Cronbach's α values for the three dimensions. The highest factor loadings were observed for item 3 (0.86) followed by item 7 (0.85), while the lowest factor loading was for item 6 (0.63). Cronbach's α values for the three dimensions were acceptable, all above 0.70 (see Table 3 and Fig 2).

### Rasch model

The Likelihood Ratio Test was significant, confirming the suitability of the PCM for the present study's data. The person separation reliabilities for the health, body function, and

**Table 1. Socio-demographic profiles of participants.**

| | | Count (%) or median (25–75) percentiles |
|---|---|---|
| Age | | 24 (21–36) |
| Gender | Female | 513 (63.5%) |
| | Male | 295 (36.5%) |
| Education level | Bachelor's degree | 501 (62%) |
| | Other | 307 (38%) |
| Income status* | Less than 500 JOD | 454 (56.2%) |
| | 500–1000 JOD | 239 (29.6%) |
| | More than 1000 JOD | 115 (14.2%) |
| Marital status | Married | 244 (30.2%) |
| | Not married | 564 (69.8%) |
| Residential Area | Urban | 580 (71.8%) |
| | Rural | 228 (28.2%) |
| Smoking status | Smoker/Ex-smoker | 201 (24.9%) |
| | Never smoked | 607 (75.1%) |
| Medical insurance | No | 233 (28.8%) |
| | Yes | 575 (71.2%) |
| Physical activity | Sedentary lifestyle | 506 (62.6%) |
| | ≤ 3 times/week | 243 (30.1%) |
| | ≥ 4 times/week | 59 (7.3%) |
| Self-rated eating habits | Healthy | 285 (35.3%) |
| | Unhealthy | 523 (64.7%) |
| Chronic diseases | No | 650 (80.4%) |
| | Yes | 158 (19.6%) |
| Medical conditions | Category 1: healthy, no comorbidities | 520 (64.4%) |
| | Category 2: comorbidities, no hospitalization | 148 (18.3%) |
| | Category 3: comorbidities, hospitalized | 84 (10.4%) |
| | Category 4: comorbidities, hospitalized ≥ 3 times | 33 (4.1%) |
| | Category 5: dependent on medicine and/or medical procedure(s) and/or medical equipment to keep alive | 23 (2.8%) |

*1 JOD is equivalent to 1.4 USD.

perception dimensions were 0.87, 0.84 and 0.66, respectively, while the item separation reliabilities for the three dimensions were 0.91, 0.91, and 0.85, respectively.

Table 4 displays the infit and outfit values and thresholds for the HQR-6D items. All items across the three dimensions had infit and outfit values within the acceptable range, confirming the model's item hierarchy and its ability to differentiate between varying levels of symptom severity among participants.

All thresholds were appropriately ordered. The lowest threshold was the first threshold for the item "I have a problem attending to my self-care needs" (−5.27) in the body function dimension, while the highest threshold was the third threshold for the item "I feel a lack of physical energy" (2.86) in the health dimension.

Fig 3 displays the Wright map for the three dimensions. The map shows that participants were distributed across all levels of disease impact on their QoL, with the majority positioned in the mid-range. The DIF analysis revealed that the difference between genders on the logit scale was minimal, at only 0.1 logits, indicating that the model was not biased by gender.

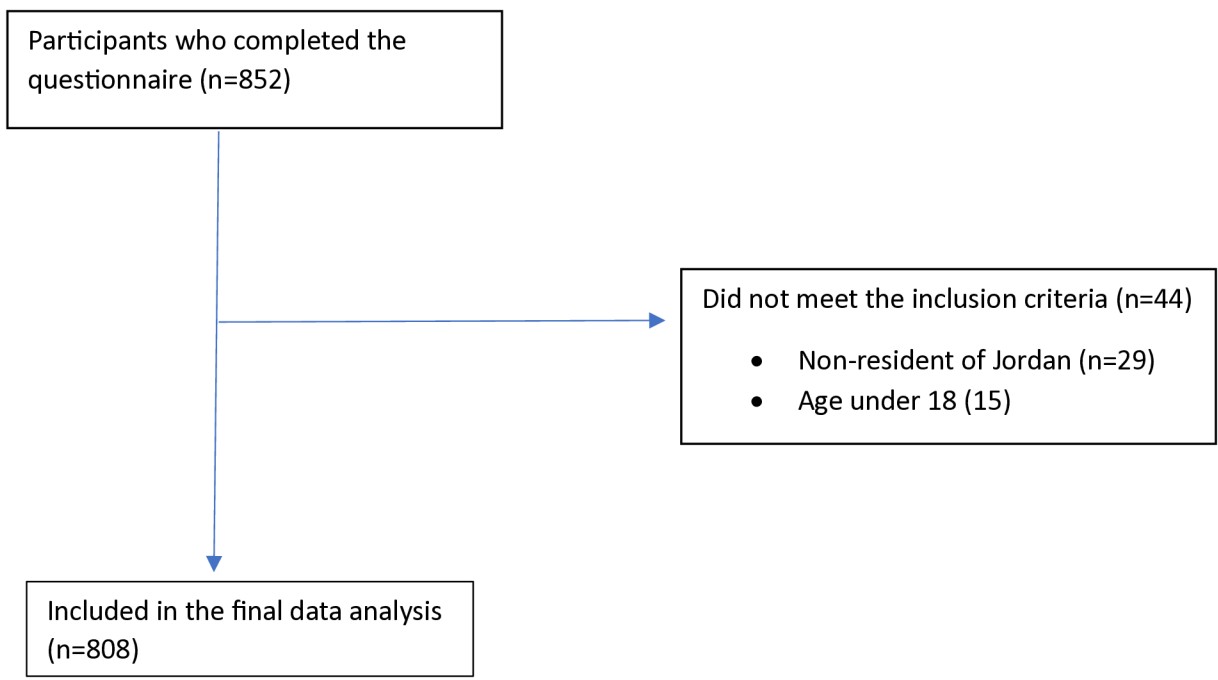

**Fig 1. Strobe flow diagram for patient enrollment.**

Table 2. Frequency of Ar-HRQ-6D items.

| Item number | Dimensions/Items | Strongly agree | Agree | Neutral | Disagree | Strongly disagree |
|---|---|---|---|---|---|---|
| | | Count (%) | | | | |
| Health dimension | | | | | | |
| Item 1 | I feel a lack of physical energy | 97 (12%) 59% | 380 (47%) | 134 (16.6%) | 137 (17%) | 60 (7.4%) |
| Item 2 | I feel tired even at rest | 83 (10.3%) 49.2% | 314 (38.9%) | 126 (15.6%) | 206 (25.5%) | 79 (9.8%) |
| Item 3 | I feel unhealthy | 75 (9.3%) 41.2% | 258 (31.9%) | 165 (20.4%) | 214 (26.5%) | 96 (11.9%) |
| Item 4 | I feel pain in any part of my body | 82 (10.1%) 43.1% | 267 (33%) | 149 (18.4%) | 215 (26.6%) | 95 (11.8%) |
| Item 5 | I feel depressed | 106 (13.1%) 38.6% | 206 (25.5%) | 185 (22.9%) | 199 (24.6%) | 112 (13.9%) |
| Item 6 | I feel anxious | 106 (13.1%) 50.7% | 304 (37.6%) | 166 (20.5%) | 166 (20.5%) | 66 (8.2%) |
| Body function dimension | | | | | | |
| Item 7 | I have difficulty to move from one place to another | 46 (5.7%) | 157 (19.4%) | 149 (18.4%) | 276 (34.2%) | 180 (22.3%) 56.5% |
| Item 8 | I have a problem attending to my self-care needs | 26 (3.2%) | 131 (16.2%) | 140 (17.3%) | 272 (33.7%) | 239 (29.6%) 63.3% |
| Item 9 | I have a problem doing household chores | 30 (3.7%) | 178 (22%) | 162 (20%) | 261 (32.3%) | 177 (21.9%) 54.2% |
| Item 10 | My movements are slower than people of my age | 29 (3.6%) | 148 (18.3%) | 158 (19.6%) | 268 (33.2%) | 205 (25.4%) 58.6% |
| Perception dimension | | | | | | |
| Item 11 | I am worried that I will suffer poor health within 5 years | 41 (5.1%) | 196 (24.3%) | 181 (22.4%) | 235 (29.1%) | 155 (19.2%) 48.3% |
| Item 12 | I am worried that my lifespan is shorter than people of my age | 21 (2.6%) | 164 (20.3%) | 167 (20.7%) | 260 (32.2%) | 196 (24.3%) 56.5% |

## Predictive validity of the Ar-HRQ-6D

The ISMT revealed significant differences in the median of the Ar-HRQ-6D scores across the different health status categories (p < 0.001) (see Fig 4). Pairwise comparisons indicated that the "healthy, no comorbidities" category had significantly higher median scores than the other

**Table 3. Confirmatory factor analysis of the HQR-6D.**

| Item | Dimensions/Items | Standardized factor loading | Standard error | Cronbach's Alpha |
|---|---|---|---|---|
| Health dimension | | | | |
| Item 1 | I feel a lack of physical energy | 0.70 | 0.00 | 0.89 |
| Item 2 | I feel tired even at rest | 0.78 | 0.05 | |
| Item 3 | I feel unhealthy | 0.86 | 0.06 | |
| Item 4 | I feel pain in any part of my body | 0.79 | 0.06 | |
| Item 5 | I feel depressed | 0.69 | 0.06 | |
| Item 6 | I feel anxious | 0.62 | 0.06 | |
| Body function dimension | | | | |
| Item 7 | I have difficulty to move from one place to another | 0.85 | 0.00 | 0.90 |
| Item 8 | I have a problem attending to my self-care needs | 0.83 | 0.03 | |
| Item 9 | I have a problem doing household chores | 0.80 | 0.03 | |
| Item 10 | My movements are slower than people of my age | 0.83 | 0.03 | |
| Perception dimension | | | | |
| Item 11 | I am worried that I will suffer poor health within 5 years | 0.85 | 0.00 | 0.81 |
| Item 12 | I am worried that my lifespan is shorter than people of my age | 0.81 | 0.04 | |

four categories, with p-values ranging from < 0.01 to 0.018. Similar findings were observed

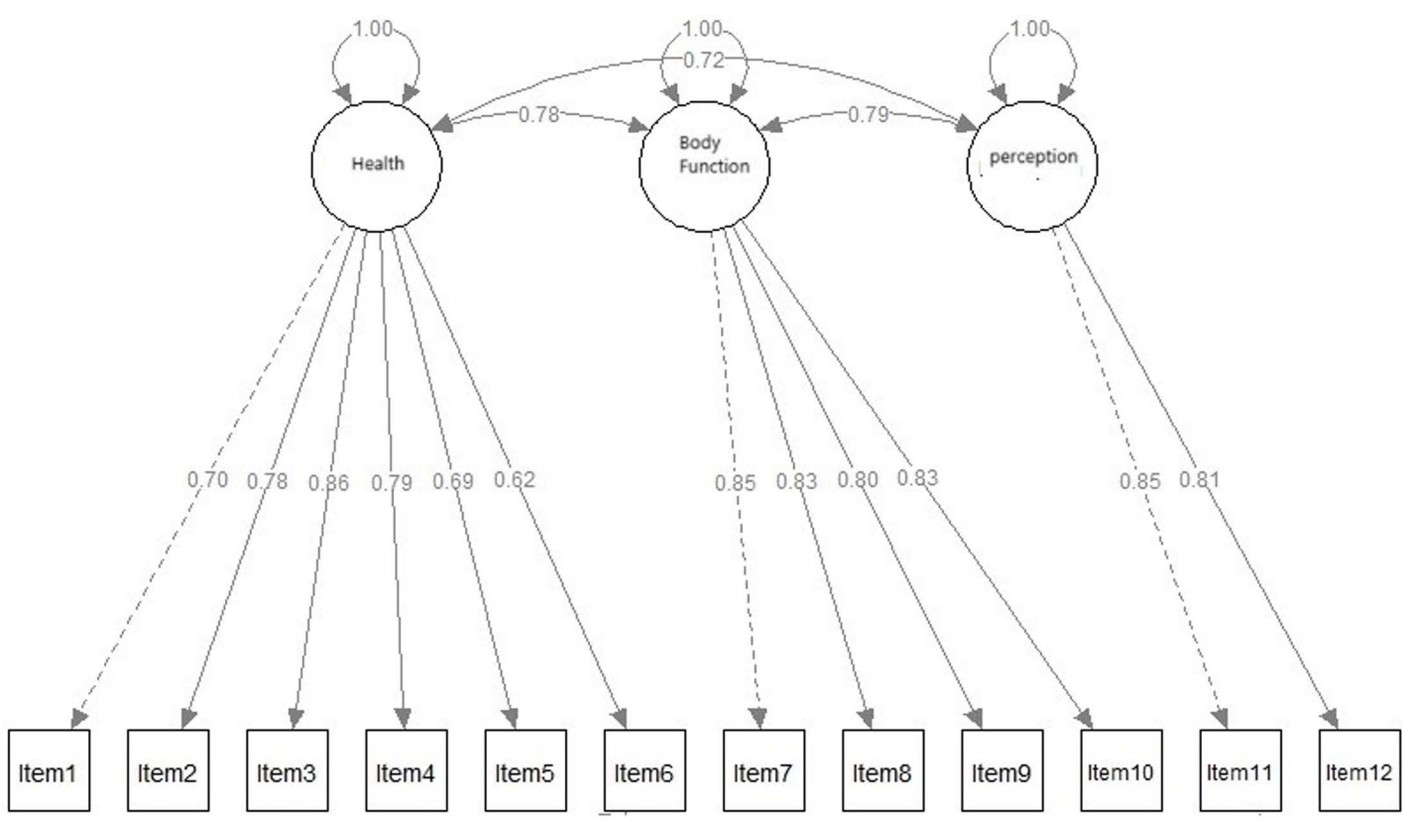

**Fig 2. Path diagram of the confirmatory factor analysis.**

Table 4. AR-HRQ-6D item outfits, infits, and thresholds.

| Items number | Dimensions/ Items | Outfit | Infit | Thresholds | | | |
|---|---|---|---|---|---|---|---|
| | | | | 1 | 2 | 3 | 4 |
| Health dimension | | | | | | | |
| Item 1 | I feel a lack of physical energy | 1.04 | 1.07 | −2.63 | 0.56 | 1.29 | 2.86 |
| Item 2 | I feel tired even at rest | 0.94 | 0.95 | −2.81 | 0.09 | 0.73 | 2.65 |
| Item 3 | I feel unhealthy | 0.80 | 0.84 | −2.90 | −0.29 | 0.59 | 2.39 |
| Item 4 | I feel pain in any part of my body | 0.99 | 0.99 | −2.77 | −0.19 | 0.59 | 2.41 |
| Item 5 | I feel depressed | 1.08 | 1.08 | −2.28 | −0.43 | 0.59 | 2.15 |
| Item 6 | I feel anxious | 1.27 | 1.23 | −2.41 | 0.16 | 1.07 | 2.81 |
| Body function dimension | | | | | | | |
| Item 7 | I have difficulty to move from one place to another | 0.93 | 0.96 | −4.37 | −1.53 | −0.37 | 1.98 |
| Item 8 | I have a problem attending to my self-care needs | 0.92 | 0.99 | −5.27 | −2.04 | −0.78 | 1.36 |
| Item 9 | I have a problem doing household chores | 1.02 | 1.09 | −5.12 | −1.49 | −0.23 | 2.00 |
| Item 10 | My movements are slower than people of my age | 0.93 | 0.99 | −5.12 | −1.82 | −0.49 | 1.70 |
| Perception dimension | | | | | | | |
| Item 11 | I am worried that I will suffer poor health within 5 years | 0.90 | 0.93 | −3.80 | −1.06 | 0.10 | 1.90 |
| Item 12 | I am worried that my lifespan is shorter than people of my age | 0.94 | 0.96 | −4.61 | −1.49 | −0.32 | 1.52 |

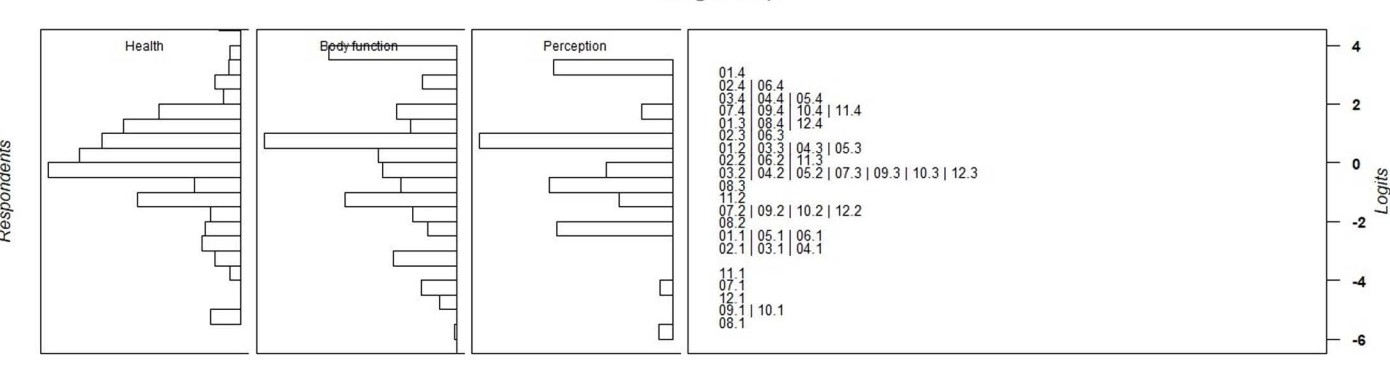

**Fig 3. Wright map of the Rasch analysis.**

across all three domains, where the "healthy, no comorbidities" category consistently had the highest median scores compared to the other categories.

## Discussion

This study validated the Arabic version of the HRQ-6D as a reliable and comprehensive tool for assessing health-related quality of life (HRQOL) in the Jordanian adult population. By addressing six key domains—pain, physical strength, emotional well-being, mobility, self-care, and future health perceptions—the HRQ-6D provides a holistic measure of HRQOL, which is essential for understanding the multifaceted impact of health on individuals' lives.

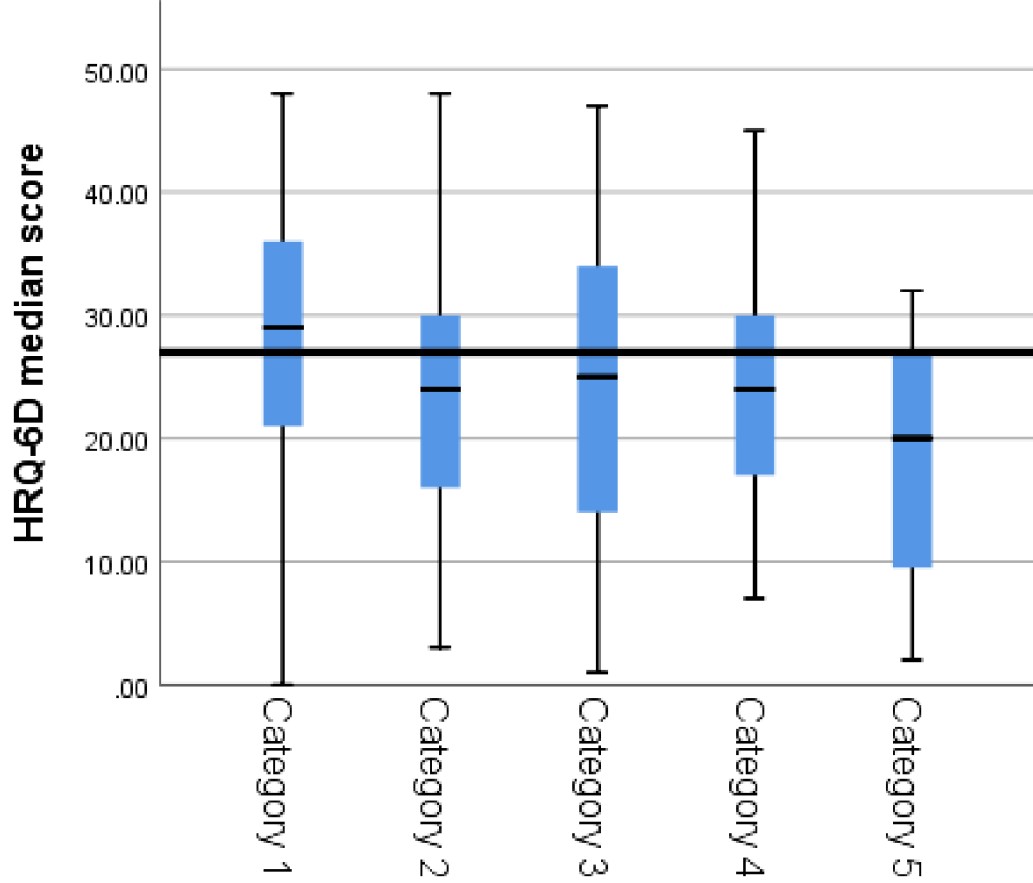

**Fig 4. HRQ6-D median scores of the different health status groups.** Category 1: healthy, no comorbidities; Category 2: comorbidities, no hospitalization; Category 3: comorbidities, hospitalized; Category 4: comorbidities, hospitalized ≥ 3 times; Category 5: dependent on medicine and/or medical procedure(s) and/or medical equipment to keep alive.

The internal consistency of the Ar-HRQ-6D, as reflected in Cronbach's alpha values for all three dimensions, is comparable to previous research, which suggests that Cronbach's alpha values above 0.70 indicate acceptable internal consistency [23]. These results confirm that the Ar-HRQ-6D's items are cohesive in measuring their respective constructs, supporting its use in assessing a wide range of HRQOL aspects. Furthermore, the CFA produced fit indices within acceptable ranges, consistent with findings from similar studies that validate multidimensional HRQOL tools [24]. The strong factor loadings for items like "I feel unhealthy" and "I have difficulty moving from one place to another" align with prior literature that emphasises the central role of physical health and mobility in HRQOL assessments [25,26]. Conversely, the lower factor loading for "I feel anxious" highlights the potential variability in how emotional aspects of health are perceived across different populations, but the value remains within acceptable limits for its inclusion as a relevant construct in the perception dimension.

The Rasch model analysis also confirmed the tool's robustness in distinguishing between different levels of HRQOL, further supporting the AR-HRQ-6D's clinical applicability. Similar to the findings by Bond [27], the person and item separation reliability values in this study demonstrate that the AR-HRQ-6D can effectively differentiate between various health states, making it a valuable instrument for identifying patients with more severe health concerns. The fit statistics, including infit and outfit values, were all within the expected range, confirming

the tool's item hierarchy and its ability to accurately reflect symptom severity among participants. These findings are consistent with prior Rasch analyses of HRQOL instruments, which highlight the importance of ensuring that health-related tools can accurately capture variations in disease impact [27].

This study's use of the HRQ-6D highlights the growing need for culturally appropriate HRQOL tools that are both reliable and flexible. While many existing tools, such as the EQ-5D, focus on more general measures of health status [11], the HRQ-6D captures a broader range of domains, including future health perceptions, which are often neglected in standard HRQOL assessments. This makes it particularly suited for settings where future health concerns, such as those related to chronic conditions or socioeconomic factors, are prominent. Moreover, the minimal gender bias found in the DIF analysis supports the Ar-HRQ-6D's applicability across diverse demographic groups, similar to findings in studies that tested other HRQOL tools for bias [22].

## Implications for clinical practice

The findings of this study highlight the Ar-HRQ-6D as a culturally appropriate and comprehensive tool for assessing HRQOL in Arabic-speaking populations. Its six domains enable a holistic evaluation of patients' health, guiding personalised care by identifying areas of difficulty, such as pain, mobility, or emotional well-being. This can help clinicians prioritise interventions and tailor treatment plans based on individual needs.

The minimal gender bias observed enhances the tool's applicability across diverse populations, ensuring equitable assessments in clinical practice. Moreover, its predictive validity makes it valuable for tracking changes in HRQOL over time, particularly in managing chronic conditions where monitoring outcomes is crucial for effective care. By integrating the Ar-HRQ-6D into routine assessments, healthcare providers could improve resource allocation and deliver more patient-centred care.

## Strengths and limitations

This study is strengthened by its rigorous methodological approach, which included the use of CFA and Rasch analysis. These techniques provided robust evidence of the tool's reliability, construct validity, and ability to differentiate between varying levels of health-related quality of life. Additionally, the large sample size and inclusion of participants from diverse locations across Jordan enhance the generalizability of the findings, particularly for young adults in Arabic-speaking populations. Additionally, the minimal gender bias observed in this study aligns with prior research on HRQOL measures, further supporting the AR-HRQ-6D's adaptability across genders and demographic groups.

Despite these strengths, the study has some limitations. The sample was primarily young and unmarried, which may limit the generalisability of the findings to other populations. Additionally, the cross-sectional design of the study does not allow for an assessment of changes in HRQOL over time. Future research should explore the performance of the Ar-HRQ-6D in more diverse populations, including individuals with chronic health conditions. Additionally, test-retest reliability should be evaluated, and longitudinal studies conducted to better understand how HRQOL evolves in response to various health interventions or life events. Furthermore, extending the validation of the Ar-HRQ-6D across different cultural and socioeconomic contexts would enhance its applicability, similar to the efforts made with existing tools like the EQ-5D. Finally, as Google Forms was used to collect the data, it was not possible to ensure the absence of duplicate responses. Implementing measures to eliminate potential duplications could have compromised the confidentiality of the study.

## Conclusions

The Ar-HRQ-6D has been validated as a reliable and comprehensive tool for assessing health-related quality of life among Jordanian adults. Its capacity to capture a broad spectrum of health perceptions, along with its minimal gender bias, makes it a valuable instrument for both clinical and research purposes. The Ar-HRQ-6D's strong ability to differentiate between varying levels of symptom severity highlights its potential to inform health interventions. Future research should further investigate its applicability across diverse populations and settings to ensure its effectiveness in enhancing health outcomes.

## Author contributions

**Conceptualization:** Abdel Qader Al Bawab, Fawaz Alasmari, Raghd Obidat.

**Data curation:** Walid Al-Qerem, Alaa Hammad, Lujain Al-Sa'di.

**Formal analysis:** Walid Al-Qerem, Lujain Al-Sa'di.

**Investigation:** Lujain Al-Sa'di, Raghd Obidat, Sarah Abu Hour, Taha Al-Hayali.

**Methodology:** Fawaz Alasmari.

**Supervision:** Anan Jarab, Judith Eberhardt.

**Writing – original draft:** Walid Al-Qerem, Anan Jarab, Abdel Qader Al Bawab, Judith Eberhardt, Fawaz Alasmari, Alaa Hammad, Lujain Al-Sa'di, Raghd Obidat, Sarah Abu Hour, Taha Al-Hayali.

**Writing – review & editing:** Anan Jarab, Abdel Qader Al Bawab, Judith Eberhardt, Fawaz Alasmari, Alaa Hammad, Lujain Al-Sa'di, Raghd Obidat, Sarah Abu Hour, Taha Al-Hayali.

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
