## [Decision Letter · Decision Letter 0]

19 Jan 2025

PONE-D-24-49309Validation and adaptation of The Arabic version of Health-Related Quality of Life with Six Domains (HRQ-6D): A Factor and Rasch analyses studyPLOS ONE

Dear Dr. Al-Qerem, Thank you for submitting your manuscript to PLOS ONE. After careful consideration, we feel that it has merit but does not fully meet PLOS ONE’s publication criteria as it currently stands. Therefore, we invite you to submit a revised version of the manuscript that addresses the points raised during the review process.

Please submit your revised manuscript by Mar 05 2025 11:59PM. If you will need more time than this to complete your revisions, please reply to this message or contact the journal office at plosone@plos.org . Please include the following items when submitting your revised manuscript:

We look forward to receiving your revised manuscript.

Kind regards,

Othman A. Alfuqaha, Ph.D.

Academic Editor

PLOS ONE

Journal Requirements:

Additional Editor Comments:

Dear Authors,

I would like to express my sincere gratitude for your valuable contribution to knowledge through this paper. I truly enjoyed reading it. The paper is well-written and well-structured. However, I have a few minor comments that could strengthen the paper further.

Method Section: Please specify the study design more clearly, and explain how the data collection was carried out. I believe it follows a snowball procedure, but this should be clarified. Additionally, how did you confirm that participants did not complete the survey multiple times? Further clarification here would be helpful.

Results Section: It would be beneficial to include a figure related to the Confirmatory Factor Analysis (CFA) to support your findings visually.

Table Clarification and abbreviation: In the tables, the column "Dinar" seems to be equivalent to the dollar??. Please clarify this for a broader audience, as it will improve the accessibility of the paper for international readers.

Discussion Section: More detailed discussion is needed regarding comparisons with other similar scales in the literature. This would provide a stronger context for your findings and further validate the robustness of your instrument.

Overall, the paper is excellent, and with these minor revisions, it will be even stronger.

Best regards,

Dr. Othman Alfuqaha

Reviewers' comments:

Reviewer's Responses to Questions

**Comments to the Author**

1. Is the manuscript technically sound, and do the data support the conclusions?

Reviewer #1: Yes

Reviewer #2: Yes

2. Has the statistical analysis been performed appropriately and rigorously? 

Reviewer #1: Yes

Reviewer #2: Yes

3. Have the authors made all data underlying the findings in their manuscript fully available?

Reviewer #1: Yes

Reviewer #2: Yes

4. Is the manuscript presented in an intelligible fashion and written in standard English?

Reviewer #1: Yes

Reviewer #2: Yes

5. Review Comments to the Author

Reviewer #1: Abstract

Add the abbreviation of Ar-HRQ-6D

Add definition for all abbreviation

Rewrite the conclusion better to include the objective of the study

IntroductionC

Consider describing the limitations of other HRQO instruments, and the advantage of this tool over the EQ-5D-5L which has been validated in Arabic, as a rational for investigating the EQ-5D in this study.

Material and method

Move the sample size calculation section to be after the study design and participants

Include more specific information about the pilot study. For example, what were the demographics of the 20 participants who took part in the pilot study? How were they selected, and did they differ from the target study population?

Discussion

Elaborate on the implications of the study findings on clinical practice.

Minor

Please check the language, typos and capitalization at the beginning of the sentence,

Reviewer #2: The authors validated a tool to assess HrQoL in Arabic. The study is well written, and the validation process was comprehensive and applied advance statistical methods. Nevertheless, I have few minor comments

Abstract

Explicitly state the actionable implications derived from the findings in the conclusion

Method state how this questionnaire was circulated. Inclusion/exclusion criteria

Introduction

What are the challenges faced by the public to have a good HRQOL in Jordan?

Elaborate more on the current medical practice in Jordan, two brief sentences with relevant references.

Please make sure all abbreviations are defined the first time they are used, i.e EQ-5D

Methods:

Write “multinomial” instead of multinominal

Did you conduct test-retest reliability?

How long did it take to complete each questionnaire?

Please only use a single term either survey or questionnaire not both together.

Please revise the punctuation of the manuscript; a lot of periods in wrong places.

Results

Mention the reason 44 participants were excluded.

Discussion

Consider including a one more statement that highlights the strengths of the study to balance the discussion of limitations.

The author acknowledges the study's limitations, including the cross-sectional design, which limits causal inference, and the lack of diversity in the sociodemographic characteristics of the participants., which may affect sample representation. This demonstrates an awareness of the study's limitations and a commitment to transparency in reporting.

Mention the general application for this tool

English and grammar: Although the manuscript is well written in standard English, there are several errors and typos in different sections. The authors should consider proof-reading in order to address the errors.

6. PLOS authors have the option to publish the peer review history of their article (what does this mean? ). If published, this will include your full peer review and any attached files.

**Do you want your identity to be public for this peer review?** For information about this choice, including consent withdrawal, please see our Privacy Policy .

Reviewer #1: No

Reviewer #2: **Yes: ** Dr.Bushra AbdelHadi

---

## [Author Response · Author response to Decision Letter 0]

28 Jan 2025

Editor

I would like to express my sincere gratitude for your valuable contribution to knowledge through this paper. I truly enjoyed reading it. The paper is well-written and well-structured. However, I have a few minor comments that could strengthen the paper further.

-Dear editor, Thank you for your encouragement, time and comments that significantly improved the quality of manuscript.

Method Section: Please specify the study design more clearly, and explain how the data collection was carried out. I believe it follows a snowball procedure, but this should be clarified.

-Many thanks, the following was added to the method section:” In this cross-sectional study, an online questionnaire was distributed using Google Forms via multiple Jordanian social media platforms including WhatsApp, Facebook and LinkedIn, in the period between June and September 2024”. Also, the following was added: “Convenience and snowball sampling were used to recruit participants.”

Additionally, how did you confirm that participants did not complete the survey multiple times? Further clarification here would be helpful.

The following was added to the limitation section: ”Finally, as Google Forms was used to collect the data, it was not possible to ensure the absence of duplicate responses. Implementing measures to eliminate potential duplications could have compromised the confidentiality of the study.”

Results Section: It would be beneficial to include a figure related to the Confirmatory Factor Analysis (CFA) to support your findings visually.

-A path diagram for CFA was added as suggested.

Table Clarification and abbreviation: In the tables, the column "Dinar" seems to be equivalent to the dollar??. Please clarify this for a broader audience, as it will improve the accessibility of the paper for international readers.

- A footnote was added below Table 1 to clarify that 1 JOD is equivalent to 1.4 USD.

Discussion Section: More detailed discussion is needed regarding comparisons with other similar scales in the literature. This would provide a stronger context for your findings and further validate the robustness of your instrument.

“One of the most widely used tools for assessing HRQOL is EQ-5D-3L, a standardized tool that measures an individual's overall health status [9]. The EQ-5D-3L tool consists of two parts: a descriptive section and an evaluation of the individual’s overall health status. The descriptive part covers five main domains: mobility, usual activities, self-care, anxiety/depression, and pain/discomfort. The second part assesses overall health using a visual analogue scale (EQ-VAS) ranging from zero (representing the worst imaginable health) to 100 (representing the best imaginable health) [10]. Nevertheless, various studies have recommended the replacement of the 3-level response scale with a 5-level response scale to improve the accuracy of evaluating specific health outcomes; thus, the EQ-5D-5L was developed [11,12]. However, a major drawback of the EQ-5D-5L and EQ-5D-3L is that each domain is measured by a single item only, which makes it impossible to assess the internal consistency of each domain and casts doubt on its reliability. Furthermore, the validation of the EQ-5D relied solely on expert panel opinions and comparisons of results from individuals with different health statuses [13,14]. However, it did not incorporate more advanced validation techniques, such as the Rasch model, exploratory factor analysis (EFA), or confirmatory factor analysis (CFA). Moreover, the content validity of the EQ-5D could be improved by including an additional domain to assess an individual’s current health status [15]. To address these limitations, the newly developed “Health-Related Quality of Life with Six Domains” (HRQ-6D [15] was specifically designed.

The present study sought to address the need for a comprehensive and culturally relevant tool to assess HRQOL in Arabic-speaking populations. The HRQ-6D was selected for its ability to evaluate a broader range of HRQOL dimensions with enhanced accuracy. This study aimed to validate the Arabic version of the HRQ-6D (AR-HRQ-6D), ensuring it is both reliable and culturally appropriate for use in Arabic-speaking healthcare settings.”

Overall, the paper is excellent, and with these minor revisions, it will be even stronger.

-Many thanks for your encouragement

Best regards,

Dr. Othman Alfuqaha

- Reviewer #1: Abstract

Add the abbreviation of Ar-HRQ-6D

- The following was added :” the Arabic version of the Health-Related Quality of Life with Six Domains (Ar-HRQ-6D) scale.”

Add definition for all abbreviation

- -The following was added: Health-related quality of life (HRQOL)”

Rewrite the conclusion better to include the objective of the study

- The following was added:” The study confirmed the reliability, validity, and predictive accuracy of the Arabic version of the AR-HRQ-6D. This tool is suitable for assessing patients' HRQOL across various medical settings.”

- Introduction

Consider describing the limitations of other HRQO instruments, and the advantage of this tool over the EQ-5D-5L which has been validated in Arabic, as a rational for investigating the EQ-5D in this study.

“One of the most widely used tools for assessing HRQOL is the European Quality of Life 5 Dimensions 3 Level Version (EQ-5D-3L), a standardized tool that measures an individual's overall health status [9]. The EQ-5D-3L consists of two parts: a descriptive section and an evaluation of the individual’s overall health status. The descriptive part covers five main domains: mobility, usual activities, self-care, anxiety/depression, and pain/discomfort. The second part assesses overall health using a visual analogue scale (EQ-VAS) ranging from zero (representing the worst imaginable health) to 100 (representing the best imaginable health) [10]. Nevertheless, various studies have recommended the replacement of the 3-level response scale with a 5-level response scale to improve the accuracy of evaluating specific health outcomes; thus, the 5-level version of the scale (EQ-5D-5L) was developed [11,12]. However, a major drawback of both the EQ-5D-5L and EQ-5D-3L is that each domain is measured by a single item only, which makes it impossible to assess the internal consistency of each domain and casts doubt on its reliability. Furthermore, the validation of the EQ-5D relied solely on expert panel opinions and comparisons of results from individuals with different health statuses [13,14]. However, it did not incorporate more advanced validation techniques, such as the Rasch model, exploratory factor analysis (EFA), or confirmatory factor analysis (CFA). Moreover, the content validity of the EQ-5D could be improved by including an additional domain to assess an individual’s current health status [15]. To address these limitations, the newly developed Health-Related Quality of Life with Six Domains (HRQ-6D) [15] was specifically designed.

The present study sought to address the need for a comprehensive and culturally relevant tool to assess HRQOL in Arabic-speaking populations. The HRQ-6D was selected for its ability to evaluate a broader range of HRQOL dimensions with enhanced accuracy. This study aimed to validate the Arabic version of the HRQ-6D (Ar-HRQ-6D), ensuring it is both reliable and culturally appropriate for use in Arabic-speaking healthcare settings.”

Material and method

Move the sample size calculation section to be after the study design and participants

-This was done, as suggested.

Include more specific information about the pilot study. For example, what were the demographics of the 20 participants who took part in the pilot study? How were they selected, and did they differ from the target study population?

Many thanks for your comment. This was included in the method section: “A pilot study was conducted with 30 randomly selected Jordanian participants (17 of whom were female) to ensure the clarity and suitability of the questionnaire for the target population. The median age of the 30 participants was 27 years. The researchers approached participants at various pharmacies across different locations in Jordan.”

Discussion

Elaborate on the implications of the study findings on clinical practice.

- Thank you for your comment. The following has been added to the Discussion:

“Implications for Clinical Practice

The findings of this study highlight the Ar-HRQ-6D as a culturally appropriate and comprehensive tool for assessing HRQOL in Arabic-speaking populations. Its six domains enable a holistic evaluation of patients' health, guiding personalised care by identifying areas of difficulty, such as pain, mobility, or emotional well-being. This can help clinicians prioritise interventions and tailor treatment plans based on individual needs.

The minimal gender bias observed enhances the tool's applicability across diverse populations, ensuring equitable assessments in clinical practice. Moreover, its predictive validity makes it valuable for tracking changes in HRQOL over time, particularly in managing chronic conditions where monitoring outcomes is crucial for effective care. By integrating the Ar-HRQ-6D into routine assessments, healthcare providers could improve resource allocation and deliver more patient-centred care.”

Minor

Please check the language, typos and capitalization at the beginning of the sentence,

- Thank you for your comment. The manuscript has been thoroughly proofread and edited to eliminate grammatical errors, typos and capitalization errors.

Reviewer 2

Reviewer #2: The authors validated a tool to assess HrQoL in Arabic. The study is well written, and the validation process was comprehensive and applied advance statistical methods. Nevertheless, I have few minor comments

Abstract

Explicitly state the actionable implications derived from the findings in the conclusion

-The following was added: “The study confirmed the reliability, validity, and predictive accuracy of the Arabic version of the Ar-HRQ-6D. This tool is suitable for assessing patients' HRQOL across various medical settings.”

Method state how this questionnaire was circulated. Inclusion/exclusion criteria

The following was added: “This cross-sectional study utilized an online questionnaire targeting adult Jordanians…”

Introduction

What are the challenges faced by the public to have a good HRQOL in Jordan?

-Thank you for your comment. The following was added: “Similarly, a cross-sectional study of heart failure patients in Jordan found that physical symptoms such as edema, dyspnea, activity intolerance, and fatigue significantly contributed to a decline in HRQOL [7]. Furthermore, a study conducted with diabetic patients in Jordan, revealed that insulin administration, low-income status, marital status, and presence of diabetic complications significantly influenced their quality of life.[8]”

Elaborate more on the current medical practice in Jordan, two brief sentences with relevant references.

- Thank you for your comment. The following has been added to the Introduction:

“In Jordan, healthcare services are delivered through a combination of public, private, and military institutions. The country has 122 hospitals, 70 of which are private, and a total of 16,057 hospital beds, with 51% in public hospitals. Approximately 9% of Jordan's GDP is allocated to healthcare, reflecting the country's emphasis on primary care and preventive services. However, there remains a growing need for culturally appropriate tools to evaluate HRQOL, particularly in managing chronic diseases, which account for 76% of total deaths. Diabetes alone affects 34% of Jordanians aged 25 and older.”

Please make sure all abbreviations are defined the first time they are used, i.e EQ-5D

-This has been checked and corrected where necessary.

Methods:

Write “multinomial” instead of multinominal

-This was corrected.

Did you conduct test-retest reliability?

-Thank you for your comment. The following was added to the limitations section: “Future research should explore the performance of the AR-HRQ-6D in more diverse populations, including individuals with chronic health conditions. Additionally, test-retest reliability should be evaluated, and longitudinal studies conducted to better understand how HRQOL evolves in response to various health interventions or life events.”

How long did it take to complete each questionnaire?

The following was added to the method section: ”The researchers approached participants at various pharmacies across different locations in Jordan. Participants confirmed that the questions and response options were clear, relevant, and appropriate and that completing the questionnaire required approximately 15 minutes.”

Please only use a single term either survey or questionnaire not both together.

-This was done as suggested.

Please revise the punctuation of the manuscript; a lot of periods in wrong places.

- This has been done.

Results

Mention the reason 44 participants were excluded.

- The following was added: “A total of 852 individuals completed the questionnaire. Of these, 44 were excluded for not meeting the inclusion criteria (15 were under the age of 18, and 29 were not residents of Jordan).”

Discussion

Consider including a one more statement that highlights the strengths of the study to balance the discussion of limitations.

-Thank you for your comment. The following was added: ” This study is strengthened by its rigorous methodological approach, which included the use of CFA and Rasch analysis. These techniques provided robust evidence of the tool’s reliability, construct validity, and ability to differentiate between varying levels of health-related quality of life. Additionally, the large sample size and inclusion of participants from diverse locations across Jordan enhance the generalizability of the findings, particularly for young adults in Arabic-speaking populations. Additionally, the minimal gender bias observed in this study aligns with prior research on HRQOL measures, further supporting the AR-HRQ-6D’s adaptability across genders and demographic groups.

”

The author acknowledges the study's limitations, including the cross-sectional design, which limits causal inference, and the lack of diversity in the sociodemographic characteristics of the participants., which may affect sample representation. This demonstrates an awareness of the study's limitations and a commitment to transparency in reporting.

-Thank you for your comment.

Mention the general application for this tool

-The following was added to the discussion:” This study validated the Arabic version of the HRQ-6D as a reliable and comprehensive tool for assessing health-related quality of life (HRQOL) in the Jordanian adult population. By addressing six key domains—pain, physical strength, emotional well-being, mobility, self-care, and future health perceptions—the HRQ-6D provides a holistic measure of HRQOL, which is essential for understanding the multifaceted impact of health on individuals' lives.”

English and grammar: Although the manuscript is well written in standard English, there are several errors and typos in different sections. The authors should consider proof-reading in order to address the errors.

- Thank you for your comment. The manuscript has been thoroughly proofread and edited to eliminate grammatical errors, typos, and any other errors.

---

## [Editor Report · Decision Letter 1]

30 Jan 2025

Validation and adaptation of The Arabic version of Health-Related Quality of Life with Six Domains (HRQ-6D): A Factor and Rasch analyses study

PONE-D-24-49309R1

Dear Dr.

<table border="0" cellpadding="0" cellspacing="0" class="datatable3" style="border-collapse: collapse; width: 678px; line-height: 14px; color: rgb(0, 0, 51); font-family: verdana, geneva, arial, helvetica, sans-serif; font-size: 11.2px;"> <tbody> <tr style="background-color: rgb(244, 244, 244);"> <td style="padding: 3px; border: 1px solid rgb(255, 255, 255);">Walid Al-Qerem</td> </tr> <tr style="background-color: rgb(244, 244, 244);"> <td style="padding: 3px; border: 1px solid rgb(255, 255, 255); width: 196.094px;"> </td> </tr> </tbody></table>

We’re pleased to inform you that your manuscript has been judged scientifically suitable for publication and will be formally accepted for publication once it meets all outstanding technical requirements.

Kind regards,

Othman A. Alfuqaha, Ph.D.

Academic Editor

PLOS ONE

Additional Editor Comments (optional):

Dear authors,

I would like to take this moment to thank you for your achievement.

All of reviewers and my comments have been addressed properly.

Keep up the great work!
---

## [Editor Report · Acceptance letter]

PONE-D-24-49309R1

PLOS ONE

Dear Dr. Al-Qerem,

I'm pleased to inform you that your manuscript has been deemed suitable for publication in PLOS ONE. Congratulations! Your manuscript is now being handed over to our production team.

Kind regards,

on behalf of

Dr. Othman A. Alfuqaha

Academic Editor

PLOS ONE